# Reasonable Nitrogen Fertilizer Management Improves Rice Yield and Quality under a Rapeseed/Wheat–Rice Rotation System

**Peng Ma [1], Yan Lan [2], Xu Lv [3], Ping Fan [3], Zhiyuan Yang [3], Yongjian Sun [3], Rongping Zhang [1,\*] and Jun Ma [3,\*]**

[1] School of Life Science and Engineering, Southwest University of Science and Technology, Mianyang 610000, China; Mapeng1@stu.sicau.edu.cn

[2] College of Agronomy, Sichuan Agricultural University, Wenjiang, Chengdu 611130, China; lanyan@stu.sicau.edu.cn

[3] Rice Research Institute, Sichuan Agricultural University, Wenjiang, Chengdu 611130, China; lvxu@stu.sicau.edu.cn (X.L.); 2020211006@stu.sicau.edu.cn (P.F.); 14236@sicau.edu.cn (Z.Y.); yongjians1980@sicau.edu.cn (Y.S.)

\* Correspondence: zhzhrrpp@163.com (R.Z.); majun@sicau.edu.cn (J.M.); Tel.: +86-159-8340-0681 (R.Z.); +86-136-0822-2603 (J.M.)

**Abstract:** To determine the influence of N fertilizer management on rice yield and rice quality under diversified rotations and establish a high-yield, high-quality, and environmentally friendly diversified planting technology, a rapeseed/wheat–rice rotation system for 2 successive years was implemented. In those rotation systems, a conventional N rate (Nc; 180 kg/hm$^2$ N in rape season, 150 kg/hm$^2$ N in wheat season) and a reduced N rate (Nr; 150 kg/hm$^2$ N in rape season, 120 kg/hm$^2$ N in wheat season) were applied. Based on an application rate of 150 kg/hm$^2$ N in the rice season, three N management models were applied, in which the application ratio of base:tiller:panicle fertilizer was 20%:20%:60% in treatment M1, 30%:30%:40% in treatment M2, and 40%:40%:20% in treatment M3. Zero N was used as the control (M0). The results showed that, under Nc and Nr in the rape season, M3 management produced an increase in rice yield. The average rice yields in 2018 and 2019 were 9.41 t/hm$^2$ and 9.54 t/hm$^2$, respectively. An increase in rice peak viscosity, hot viscosity, break disintegration, and chalkiness was achieved. Under Nc and Nr in the wheat season, the panicle fertilizer of 40%:40%:20% in rice season produced a higher rice yield. The average yield was 9.45 t/hm$^2$ and 9.19 t/hm$^2$, respectively, and an increase in rice peak viscosity, hot viscosity, and break disintegration was produced. Reduced N for rapeseed and the panicle fertilizer of 40%:40%:20% in rice season under a rapeseed–rice rotation system can be recommended to stabilize yield and ensure high-quality rice production and environmentally friendly rapeseed–rice rotation systems in southern China.

**Keywords:** rapeseed/wheat–rice rotation system; nitrogen management; rice yield; rice quality

## 1. Introduction

It is known that rice plays an important role in the world. With the improvement of living standards, consumers pay more attention to rice quality, particularly the eating/cooking quality. Studies have found that rice varieties with high amylose content have poor eating/cooking quality [1], while rice varieties with low amylose content generally have a higher eating/cooking quality. Quality is an important consideration in rice production. Rice quality is not only controlled by genetic factors but also affected by temperature, water, and nitrogen management. Nitrogen is an important element in fertilizer that can significantly affect the grain yield and quality of rice [2,3]. Reasonable nitrogen fertilizer management not only increases the yield of rice but is also an important cultivation measure to regulate the quality of production [4,5]. There is a large nitrogen fertilizer input in China's nitrogen-fertilized farmland, and the nitrogen utilization rate

is only about 30%. The high nitrogen fertilizer input not only reduces the use of nitrogen but also causes environmental pollution. The nitrogen fertilizer enters the water body, soil, and air, causing water and soil pollution [6,7]. Diversified crop rotation models, such as wheat–rice crop rotations and rapeseed–rice crop rotations, are widely distributed and produce a large amount of straw every year in China. Rapeseed and wheat straw contains abundant nutrients, and its incorporation has become one of the important methods used to decrease the application of N and other chemical fertilizers. Straw returned to the field can effectively increase the rice yield, roughness rate, polished and whole rice rate, reduce chalky grain rate, chalkiness and amylose content, increase the aspect ratio and gel consistency, and improve the rice processing quality and appearance quality, taste quality, and nutritional quality [8,9]. Yan et al. [10] showed that returning wheat straw to the field can increase rice yield. At the same time, it can reduce the chalky grain rate and chalkiness and improve the rice quality under a rice-wheat rotation system. The optimization of straw return to the field and nitrogen fertilizer under different rotation modes can not only realize the efficient use of resources but also effectively improve economic benefits [11]. Under the wheat–rice and rapeseed–rice rotation model, straw returned to the field and nitrogen fertilizer management have a significant effect on the nitrogen use efficiency of hybrid rice [12]. Returning straw to the field combined with on-site nitrogen fertilizer management can increase yield and improve the appearance and taste quality of rice [13]. Under the condition of a nitrogen application rate of 276 kg·hm$^{-2}$, if only high-quality rice is required, a nitrogen fertilizer operation with a ratio of base tiller to panicle fertilizer of 10:0 should be used. For high-quality rice, nitrogen fertilizer management with a 7:3 ratio between the base tiller fertilizer and spike fertilizer should be used [14]. Under conditions of a nitrogen application rate of 2.25 t/hm$^2$ and 4.50 t/hm$^2$, and a nitrogen fertilizer operation with a ratio of the base tiller to panicle fertilizer of 6:4–8:2, the chalkiness and amylose content of rice are reduced, while the gel consistency and protein content are reduced, which can improve the cooking and eating quality and nutritional quality of rice [15]. The above research mainly focuses on the effects of the straw return to the field, nitrogen fertilizer management, and the supporting nitrogen fertilizer management under a single rotation mode on rice yield and rice quality, but there are few comparative studies between straw return to the field and nitrogen fertilizer management under different rotation modes. Therefore, in this study, under a rapeseed–rice and wheat–rice rotation system, the straw of rape and wheat were returned to the field, and different nitrogen fertilizer management treatments were used. The objective was to determine the effects of optimized nitrogen fertilizer application on the yield and quality of hybrid indica rice under a rapeseed/wheat–rice rotation system. In so doing, the regulation and control methods for the quality and yield improvement of hybrid indica rice under the diversified rotation system can be identified, with a view to improving the quality of rice under different rotation models in production.

## 2. Materials and Methods

### 2.1. Experimental Site Information

The experiments were conducted at the farm of the Rice Research Institute, Sichuan Agricultural University, Wenjiang, Sichuan Province, China (30.70° N, 103.83° E) from October 2017 to early September 2019. Immediately before the field experiment (2017), soil samples from the top 0.20 m of surface soil contained 1.52 g/kg total N (Kjeldahl method, UDK-169, ITA), 23.89 mg/kg of available phosphorus (Mo–Sb colorimetry after digestion with $H_2SO_4$ and $HClO_4$), 2.421% organic matter ($K_2Cr_2O_7$-volumetric method), and 52.61 mg/kg available K (flame spectrometry after $NH_4OAc$ extraction) and had a pH of 6.19 (tested in a sample containing a 1:2.5 ratio of soil to water). The average air temperature and precipitation during the previous crop and rice-growing season, measured at the weather station close to the experimental site, are detailed in Figure 1.

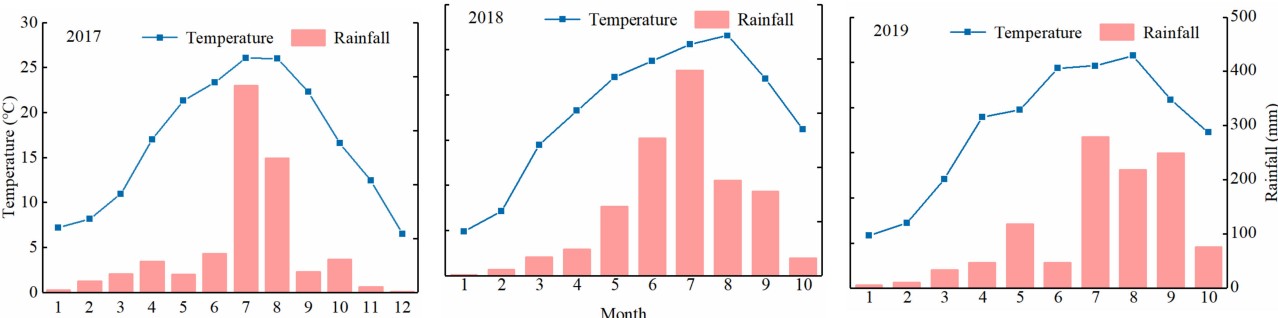

**Figure 1.** The meteorological data of the experimental area, including temperature and rain full in 2017–2019.

### 2.2. Experiment Design

The experiment adopted a three-factor design. The first factor (two levels) was the previous crop, which was rape and wheat represented by Pr and Pw, respectively. The second factor (two levels) was the N application rate in the previous season, with a conventional N application (Nc; 180 kg/hm$^2$ N in rape season, 150 kg/hm$^2$ N in wheat season), reduced N (Nr; 150 kg/hm$^2$ N in rape season, 120 kg/hm$^2$ N in wheat season), and N fertilizer as basal manure and top dressing at a 5:5 ratio. The third factor (three levels) was N fertilizer management, with common urea as the N source. Based on an application rate of 150 kg/hm$^2$ N in the rice season, three N management models were used, where the ratio of the application of base fertilizer, tiller fertilizer, and panicle fertilizer was 20%:20%:60% in treatment M1, 30%:30%:40% in treatment M2, and 40%:40%:20% in treatment M3. M0 was defined as the zero-N control. A total of 16 treatments were performed with three repetitions. Each experimental plot was 15.75 m$^2$ in area with a 30 cm-wide ridge covered with a plastic film to prevent water and nutrient penetration from the contiguous plots.

The rape variety used was 'Mianyou No. 15' (Mianyang Academy of Agricultural Sciences, Sichuan Province), and the wheat variety used was 'Shumai 969' (Wheat Research Institute of Sichuan Agricultural University). Rape seedlings were transplanted on 12 October 2017 and 2018 and spaced at $0.5 \times 0.35$ m (57,000 plants/hm$^2$) in both 2017 and 2018. The rapeseed was harvested on 1 May 2018 and 2019, and the wheat was harvested on 8 May 2018 and 2019. Straw was cut into 5 cm pieces and returned to the corresponding plots after the rape and wheat harvest. The N contribution to the plot by rape straw was 16.08–27.81 kg/hm$^2$, 13.08–19.88 kg/hm$^2$ by wheat straw. Urea (N, 46.4%) was used as the N source, phosphorus (P$_2$O$_5$, 12.0%) was used as the phosphorus source, and potassium chloride (K$_2$O, 60.0%) was used as the potassium source. Phosphorus and potassium were applied to the soil as a base fertilizer 1 day before sowing or transplanting. Nitrogen, phosphorus, and potassium fertilizers were applied in the rape season at a ratio of 2:1:2 and in the wheat season at a ratio of 2:1:1.

'Fyou 498' a commonly planted, high-yield, indica hybrid rice cultivar, was sown in a seedbed on 17 April 2018 and 2019, and seedlings were transplanted to the field on 23 May 2018 and 2019. The rice seedlings were transplanted and spaced at $0.333 \times 0.167$ m, in both 2018 and 2019, with one plant per hill. Ordinary urea was applied during the rice season. Nitrogen, phosphorus, and potassium fertilizers were applied in the rape season at a ratio of 2:1:2. Phosphate fertilizer (P$_2$O$_5$; 75 kg/hm$^2$) and potash fertilizer (K$_2$O; 150 kg/hm$^2$) were used as base fertilizers. The base fertilizer was applied 1 day before transplanting. Tiller fertilizer was applied 7 days after transplanting. Spike fertilizer, divided into flower-promoting fertilizer and flower-keeping fertilizer at a ratio of 5:5, was applied twice at the four-leaf and two-leaf stages. After the rice was harvested, the entire amount of straw was chopped and returned to the corresponding plot, and the N contribution to the plot by rape straw was 16.08–27.81 kg/hm$^2$. The weeds were controlled in the rape and rice plots with

pretilachlor (1720 mL/hm$^2$) (Jiangsu Changlong Agrochemicals Co., Ltd. Taizhou, China). The herbicide was applied once at the seedling stage of rapeseed and the tillering stage of rice. Pests and diseases were controlled by imidacloprid (90 g/hm$^2$) (Hubei Xinhe Chemical Co., Ltd. Wuhan, China) and kasugamycin (1200 mL/hm$^2$) (Hubei Dibai Chemical Co., Ltd., Wuhan, China) to avoid yield loss.

### 2.3. Measurements and Methods

2.3.1. Plant Sampling and Measurements

At the maturity stage, all rice plants were selected from each plot to test the rice yield (GY) were calculated according to the actual number of harvested plants; the value was adjusted to 13.5% moisture to ensure safe storage.

2.3.2. Rice Quality Index Measurements

Rice grains were collected, dried, and stored for more than 3 months, according to NY/T83-1988 (1988). Grain samples of 120 g with 3 replications from each plot were collected for grain quality analysis according to GB/T 17891-1999 (1999). The brown rice, milled rice, and head rice rates were expressed as percentages of the total grain weights. Chalkiness was evaluated on 100 milled grains per plot. The number of grains containing over 20% white was considered as chalkiness rate. The chalkiness size was expressed as the percentage of the total area of the kernel. The amylose content was determined from the absorption at 620 nm by scanning the iodine absorption spectrum from 400 to 900 nm using a spectrophotometer (Ultrospec 6300 pro, Amersham Biosciences, Little Chalfont, UK). The values were converted to amylose content by reference to a standard curve prepared from rice. The protein content was measured with a grain analyzer (Infratec 1241, Foss, Denmark). Rice paste properties were determined using a Rapid Visco Analyser (RVA; Super3, Newport Scientific, Sydney, Australia), following the procedure of the American Association of Cereal Chemists. Three-gram samples of flour were sifted with a 0.15 mm sieve and mixed with 25 g of deionized water in an RVA sample tube. Peak viscosity, hot viscosity, cool viscosity in centipoise units (cp), and their derivative parameter breakdown (peak viscosity minus hot viscosity), setback (cool viscosity minus peak viscosity), and consistency (cool viscosity minus hot viscosity) were recorded with matching software, Thermal Cline for Windows (TCW). Cooking/eating quality was measured by Taste Analyzer RCTA11A (Satake Co., Hiroshima, Japan). The primary function of the taste analyzer was to convert various physicochemical parameters of rice into taste value.

### 2.4. Data Analysis

Data were analyzed using analysis of variance (ANOVA), and the means were compared based on the least significant difference (LSD) test at the 0.05 probability level using SPSS23 (Chinese version v22.0.0.0) (Statistical Product and Service Solutions Inc., Chicago, IL, USA). The Origin Pro 2017(OriginLab, Northampton, MA, USA) was used to draw the figures.

## 3. Results

### 3.1. Effects of N Application Rate in the Previous Season and N Management in Rice Season, on Rice Yield

The analysis of variance showed that the previous crop (P), nitrogen application rate (N), nitrogen fertilizer management (M), and their interaction effects reached significant levels, and there were also differences between treatments in the 2 years (Table 1).

**Table 1.** Analysis of the variance of rice yield by nitrogen fertilizer management under a rapeseed/wheat–rice rotation system.

| Source of Variation | Degree of Freedom | Sum of Squares | Mean Square | Computed $F$ | $F_{0.05}$ | $F_{0.01}$ |
|---|---|---|---|---|---|---|
| Replication | 4 | 0.01 | 0.00 | 73.13 ** | 2.53 | 3.65 |
| Treatment | 30 | 169.82 | 5.66 | 116467.42 ** | 1.65 | 2.03 |
| Year (Y) | 1 | 4.31 | 4.31 | 88609.67 ** | 4.00 | 7.08 |
| Previous crop (P) | 1 | 1.19 | 1.19 | 24546.63 ** | 4.00 | 7.08 |
| N rate (N) | 1 | 0.33 | 0.33 | 6798.35 ** | 4.00 | 7.08 |
| N management (M) | 3 | 156.81 | 52.27 | 175441.77 ** | 2.76 | 4.13 |
| Y × P | 1 | 1.19 | 1.19 | 24453.13 ** | 4.00 | 7.08 |
| Y × N | 1 | 0.01 | 0.01 | 199.67 ** | 4.00 | 7.08 |
| Y × M | 3 | 4.25 | 1.42 | 29128.42 ** | 2.76 | 4.13 |
| P × N | 1 | 0.02 | 0.02 | 344.81 ** | 4.00 | 7.08 |
| P × M | 3 | 2.94 | 0.98 | 20151.89 ** | 2.76 | 4.13 |
| N × M | 3 | 0.11 | 0.04 | 774.41 ** | 2.76 | 4.13 |
| Y × P × N | 1 | 0.08 | 0.08 | 1715.07 ** | 4.00 | 7.08 |
| Y × P × M | 3 | 1.49 | 0.50 | 10198.31 ** | 2.76 | 4.13 |
| Y × N × M | 3 | 0.82 | 0.27 | 5635.96 ** | 2.76 | 4.13 |
| P × N × M | 3 | 0.35 | 0.12 | 2423.26 ** | 2.76 | 4.13 |
| Y × P × N × M | 3 | 0.23 | 0.08 | 1567.64 ** | 2.76 | 4.13 |
| Error | 60 | 0.00 | 0.00 | | | |
| Total variation | 95 | 174.14 | | | | |

Y: year; P: previous crop; N: nitrogen rate; M: nitrogen management. * and ** mean significance at the 0.05 and 0.01 probability levels, respectively. $F$: Analysis of variance.

Further analysis shows that the change in rice output between the years is basically the same (Figure 2). The yield of rice under different previous crops was rapeseed (Pr) > wheat (Pw), and Pr increased by 2.67% relative to Pw. Under different nitrogen application rates, the performance was ranked as: conventional nitrogen application (Nc) > reduction (Nr). Under different nitrogen operations, the performance was: M3 > M2 > M1 > M0, andM3 was relative to M2, M1, and M0, and increased by 1.39%, 4.61%, and 55.67%, respectively. The interaction effect of Pr and M3 nitrogen fertilizer management on seed setting rate, thousand-grain weight, and yield was significantly higher than the interaction effect of other previous crops and different nitrogen fertilizer management treatments, and the interaction effect of Nc and M3 nitrogen fertilizer management treatment had a higher impact on yield. Interaction effects of other nitrogen application rates and different nitrogen fertilizer management strategies indicated that in the rape season, the rice yield under Nc and Nr was the highest under the M3 operation, and the 2-year average yields were 9.41 t/hm$^2$ and 9.54 t/hm$^2$, respectively. Compared with Nc, the rice yield under the M3 operation in the rice season increased by 1.38% on average in 2 years. In the wheat season, both Nc and Nr had the highest rice yield under the M2 operation, and the 2-year average yields were 9.45 t/hm$^2$ and 9.19 t/hm$^2$. Compared with Nc in wheat season, rice yield under M2 operation decreased by 2.75% on average in 2 years, and the difference was not significant. This indicates that reducing nitrogen by 20% in rapeseed season, with a ratio of the application of base fertilizer, tiller fertilizer, and panicle fertilizer of 40%:40%:20% (M3) in rice season, was more conducive to increasing rice yield.

### 3.2. Effects of N Application Rate in the Previous Season and N Management in Rice Season, on Rice Quality Characteristics

The analysis of variance shows that there are significant differences among the various indicators of rice quality depending on the year, the previous crop, the amount of nitrogen applied, the proportion of nitrogen fertilizer, and the interaction between them. The interaction of the three was not significant (Table 2). It can be seen that nitrogen fertilizer management under the rapeseed/wheat–rice rotation had a greater impact on various indicators of rice quality.

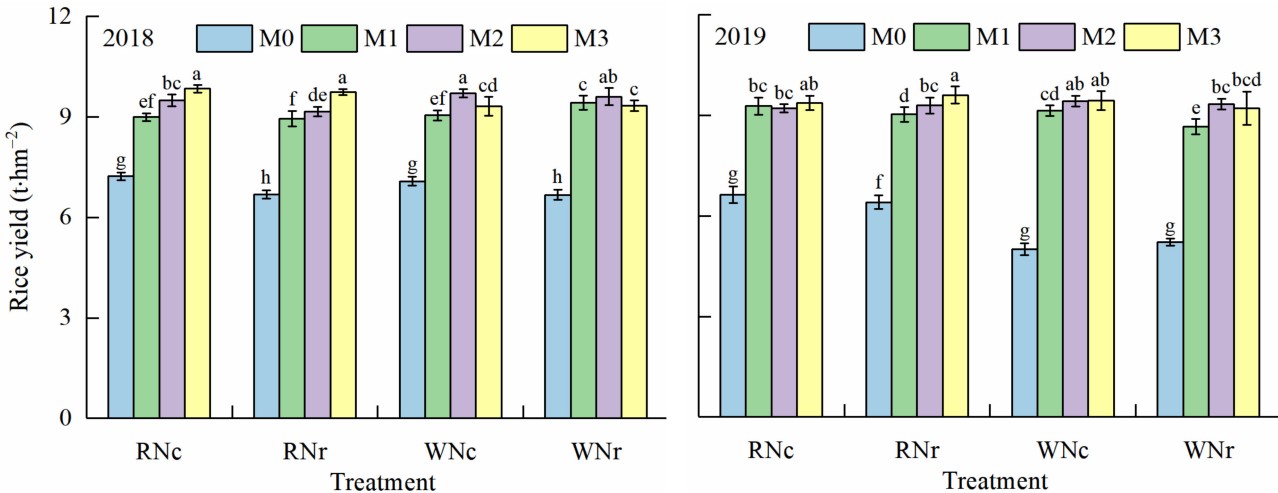

**Figure 2.** The effects of the N application rate in the previous season and N management in the rice season on the rice yield. RNc and RNr represent the conventional nitrogen application and reduced nitrogen application in the rape season, respectively. WNc and WNr represent the conventional nitrogen application and reduced nitrogen application in the wheat season, respectively. M0 represents zero N was used in rice season; M1, M2, and M3 represent based on an application rate of 150 kg/hm$^2$ N in the rice season, three N management models were applied, in which the application ratio of base:tiller:panicle fertilizer was 20%:20%:60%, 30%:30%:40%, and 40%:40%:20%, respectively. Lower case letters indicate that the yields of the hybrid rice are significantly different among the treatments ($p < 0.05$, LSD method).

### 3.2.1. Processing and Nutritional Quality

The brown rice rate, polished rice rate, and amylose content of rice are the highest in 2019 (Table 3). The brown rice rate, protein, and amylose are higher in the wheat season than in the rape season; the polished rice rate and the whole rice rate are higher in the rapeseed season than in the wheat season. Different previous crops have a significant impact on the processing and nutritional quality of rice. Brown rice rate, polished rice rate, and whole rice rate are the largest under different nitrogen application rates in rapeseed and wheat seasons under reduced nitrogen applications, while protein and amylose are the highest under conventional nitrogen application in rapeseed and wheat seasons. The brown rice rate, polished rice rate, and protein content under different nitrogen fertilizer management showed as M1 > M2 > M3 > M0; protein content increased by 4.24%, 6.97%, and 28.72% compared to M2, M3, and M0 under the treatment of M1; the content of amylose was M0 > M3 > M2 > M1 under the different nitrogen fertilizer strategies. The amylose content of rice, except for the control without nitrogen fertilizer (M0), was the highest, and all gradually decreased with the decrease of the ratio of basal tiller fertilizer. The change in protein content was opposite, with significant differences between treatments. Increasing the ratio of panicle fertilizer can improve the nutritional quality of rice.

**Table 2.** Significance of variance estimates related to years (Y), previous crop (P), nitrogen rate (N), nitrogen management (M), and their interactions on rice grain quality traits.

| Source of Variation | Year (Y) | Previous Crop (P) | N Rate (N) | N Management (M) | Y × P | Y × N | Y × M | P × N | P × M | N × M | Y × P × N | Y × P × M | Y × N × M | P × N × M | Y × P × N × M |
|---|---|---|---|---|---|---|---|---|---|---|---|---|---|---|---|
| Degree of freedom | 1 | 1 | 1 | 3 | 1 | 1 | 3 | 1 | 3 | 3 | 1 | 3 | 3 | 3 | 3 |
| BR | 5.02 * | 51.21 ** | 0.16 ns | 35.84 ** | 0.02 ns | 1.38 ns | 3.20 * | 2.23 ns | 2.48 ns | 2.32 ns | 1.20 ns | 0.24 ns | 0.69 ns | 2.44 ns | 0.42 ns |
| MR | 2.20 ns | 781.90 ** | 0.01 ns | 15.80 ** | 0.22 ns | 0.50 ns | 3.71 * | 11.20 ** | 5.31 ** | 1.84 ns | 0.05 ns | 4.06 * | 1.93 ns | 1.19 ns | 2.18 ns |
| HMR | 6.47 * | 1204.87 ** | 15.72 ** | 109.73 ** | 14.91 ** | 28.00 ** | 45.54 ** | 19.50 ** | 111.92 ** | 21.43 ** | 4.80 * | 36.51 ** | 9.16 ** | 4.12 * | 4.56 ** |
| CP | 16.43 ** | 1218.23 ** | 1.02 ns | 1202.13 ** | 90.00 ** | 196.23 ** | 24.70 ** | 2.73 ns | 22.47 ** | 33.47 ** | 49.44 ** | 30.64 ** | 14.60** | 1.38 ns | 59.22 ** |
| CD | 2.21 ns | 748.80 ** | 0.17 ns | 392.30 ** | 49.82 ** | 87.85 ** | 6.84 ** | 8.82 ** | 1.45 ns | 17.37 ** | 18.17 ** | 8.64 ** | 6.62 ** | 0.80 ns | 27.13 ** |
| AC | 661.61 ** | 613.48 ** | 4.70 * | 0.42 ns | 261.03 ** | 1.47 ns | 11.51 ** | 0.76 ns | 3.62 * | 4.65 ** | 0.76 ns | 3.62 * | 1.08 ns | 1.62 ns | 1.62 ns |
| PC | 198.58 ** | 1473.91 ** | 4.23 * | 1341.81 ** | 9.28 ** | 159.25 ** | 132.13 ** | 0.01 ns | 0.01 ns | 53.73 ** | 0.01 ns | 0.01 ns | 57.12 ** | 0.01 ns | 0.02 ns |
| Mouthfeel | 10.48 ** | 523.49 ** | 9.82 ** | 102.31 ** | 15.76 ** | 1.32 ns | 1.05 ns | 2.45 ns | 0.41 ns | 0.78 ns | 0.39 ns | 1.31 ns | 3.12 * | 10.48 ** | 8.90 ** |
| Comprehensive | 12.53 ** | 448.13 ** | 10.49 ** | 98.40 ** | 9.54 ** | 2.50 ns | 2.24 ns | 0.05 ns | 7.90 ** | 1.23 ns | 4.17 * | 2.51 ns | 2.92 * | 6.28 ** | 8.55 ** |
| PV | 8.99 ** | 1396.57 ** | 69.67 ** | 92.61 ** | 0.38 ns | 63.88 ** | 6.30 ** | 102.62 ** | 2.43 ns | 97.07 ** | 34.60 ** | 46.77 ** | 44.78 ** | 76.85 ** | 44.31 ** |
| HV | 29.38 ** | 153.92 ** | 16.62 ** | 57.96 ** | 11.28 ** | 0.01 ns | 31.43 ** | 25.64 ** | 7.29 ** | 4.04 * | 3.12 ns | 18.84 ** | 88.48 ** | 69.24 ** | 5.09 ** |
| BD | 1.03 ns | 205.19 ** | 0.39 ns | 8.10 ** | 33.87 ** | 1.01 ns | 4.15 ** | 6.37 * | 4.38 ** | 7.07 ** | 4.16 * | 6.22 ** | 3.83 * | 4.17 ** | 11.93 ** |
| CV | 5.56 * | 291.82 ** | 24.30 ** | 66.79 ** | 0.06 ns | 30.69 ** | 0.70 ns | 40.89 ** | 7.14 ** | 25.54 ** | 3.12 ns | 12.47 ** | 32.59 ** | 28.21 ** | 9.35 ** |
| SB | 1.47 ns | 1126.67 ** | 7.46 ** | 50.35 ** | 1.26 ns | 5.30 * | 81.42 ** | 0.90 ns | 7.20 ** | 5.59 ** | 1.84 ns | 15.59 ** | 17.65 ** | 4.17 ** | 18.49 ** |
| PT | 1.97 ns | 84.40 ** | 0.01 ns | 0.11 ns | 0.73 ns | 0.40 ns | 0.48 ns | 0.82 ns | 1.95 ns | 2.16 ns | 0.08 ns | 0.24 ns | 1.31 ns | 0.87 ns | 3.85 * |
| $F_{0.05}$ | 4.01 | 4.01 | 4.00 | 2.76 | 4.00 | 4.00 | 2.76 | 4.00 | 2.76 | 2.76 | 4.00 | 2.76 | 2.76 | 2.76 | 2.76 |
| $F_{0.01}$ | 7.07 | 7.08 | 7.08 | 4.13 | 7.08 | 7.08 | 4.13 | 7.08 | 4.13 | 4.13 | 7.08 | 4.13 | 4.13 | 4.13 | 4.13 |

Y: year; P: previous crop; N: nitrogen rate; M: nitrogen management; BR: brown rice rate; MR: milled rice rate; HMR: head rice rate; CP: chalk grain rate; CD: chalkiness degree; AC: amylose content; PC: protein content; PV: peak viscosity; HV: hot viscosity; BD: break disintegration; CV: cool viscosity; SB: setback; PT: peak time. * and ** mean significance at the 0.05 and 0.01 probability levels, respectively.

**Table 3.** The effects of the N application rate in the previous season and N management in the rice season on rice quality characteristics.

| Year | Treatment | | | BR (%) | MR (%) | HMR (%) | PC (%) | AC (%) |
|---|---|---|---|---|---|---|---|---|
| 2018 | Pr | Nc | M0 | 79.18 l | 65.51 de | 52.00 jkl | 4.22 q | 19.63 g |
| | | | M1 | 81.42 bc | 67.90 abc | 59.50 bcd | 5.85 h | 16.31 j |
| | | | M2 | 80.76 gh | 68.48 ab | 58.60 ef | 5.63 ij | 17.78 hi |
| | | | M3 | 81.14 cd | 68.32 abc | 58.44 ef | 5.61 ij | 18.57 h |
| | | | average | 80.63 | 67.55 | 57.14 | 5.33 | 18.07 |
| | | Nr | M0 | 79.48 l | 64.62 ef | 51.02 mn | 4.33 q | 17.85 hi |
| | | | M1 | 81.18 cd | 68.99 a | 61.38 a | 6.05 g | 16.90 ij |
| | | | M2 | 80.84 fg | 69.54 a | 60.70 abc | 5.87 h | 17.48 i |
| | | | M3 | 81.50 bc | 68.96 a | 60.18 abc | 5.75 hi | 17.69 hi |
| | | | average | 80.75 | 68.03 | 58.32 | 5.50 | 17.48 |
| | Pw | Nc | M0 | 80.03 k | 63.10 fgh | 52.23 jkl | 4.90 mn | 22.78 cd |
| | | | M1 | 81.82 abc | 63.02 fgh | 44.94 o | 6.53 bc | 22.39 de |
| | | | M2 | 81.96 abc | 63.15 fgh | 54.883 h | 6.31 def | 22.55 de |
| | | | M3 | 81.52 bc | 62.36 ghi | 54.21 hi | 6.29 ef | 22.59 de |
| | | | average | 81.33 | 62.91 | 51.57 | 6.01 | 22.58 |
| | | Nr | M0 | 80.71 gh | 61.13 i | 52.05 jkl | 5.01 m | 22.60 de |
| | | | M1 | 82.28 ab | 62.68 ghi | 51.08 lm | 6.73 a | 21.97 f |
| | | | M2 | 81.71 abc | 61.78 hi | 57.34 fg | 6.55 b | 22.47 de |
| | | | M3 | 81.48 bc | 63.19 fgh | 53.53 hij | 6.43 bcd | 22.51 de |
| | | | average | 81.55 | 62.20 | 53.50 | 6.18 | 22.39 |
| 2019 | Pr | Nc | M0 | 80.18 jk | 66.62 cd | 56.83 g | 4.60 p | 23.03 abc |
| | | | M1 | 81.61 abc | 69.16 a | 60.74 abc | 5.64 ij | 22.30 ef |
| | | | M2 | 81.35 bc | 68.05 abc | 61.00 ab | 5.59 j | 22.43 de |
| | | | M3 | 81.23 cd | 67.16 bc | 59.34 cd | 5.58 j | 22.83 bc |
| | | | average | 81.09 | 67.75 | 59.48 | 5.35 | 22.65 |
| | | Nr | M0 | 80.29 ij | 68.12 abc | 54.52 h | 4.68 op | 23.04 abc |
| | | | M1 | 81.08 de | 69.20 a | 60.82 abc | 5.83 h | 21.99 f |
| | | | M2 | 80.57 hi | 68.46 ab | 58.00 efg | 5.16 l | 22.23 ef |
| | | | M3 | 80.98 ef | 68.54 ab | 59.22 de | 4.78 no | 22.89 bc |
| | | | average | 80.73 | 68.58 | 58.14 | 5.11 | 22.54 |
| | Pw | Nc | M0 | 80.71 gh | 62.66 ghi | 52.56 jkl | 5.18 l | 24.02 a |
| | | | M1 | 81.56 bc | 62.46 ghi | 51.79 kl | 6.22 f | 23.29 abc |
| | | | M2 | 82.17 ab | 63.75 fg | 52.96 ijk | 6.17 fg | 23.42 abc |
| | | | M3 | 81.95 abc | 63.15 fgh | 50.14 n | 6.16 fg | 23.82 abc |
| | | | average | 81.60 | 63.01 | 51.86 | 5.93 | 23.64 |
| | | Nr | M0 | 81.36 bc | 61.86 hi | 51.62 kl | 5.26 kl | 24.03 a |
| | | | M1 | 82.60 a | 63.58 fgh | 54.32 hi | 6.41 cde | 22.98 abc |
| | | | M2 | 82.08 ab | 62.42 ghi | 52.44 jkl | 5.74 hi | 23.22 abc |
| | | | M3 | 81.13 cd | 62.06 ghi | 52.64 jkl | 5.36 k | 23.88 ab |
| | | | average | 81.79 | 62.48 | 52.76 | 5.69 | 23.53 |

Pr represents rapeseed; Nc and Nr represent the conventional nitrogen application and reduced nitrogen application in the rape season, respectively; Pw represents wheat; Nc and Nr represent the conventional nitrogen application and reduced nitrogen application in the wheat season, respectively. M0 represents zero N was used in rice season; M1, M2, and M3 represent based on an application rate of 150 kg/hm2 N in the rice season, three N management models were applied, in which the application ratio of base:tiller:panicle fertilizer was 20%:20%:60%, 30%:30%:40%, and 40%:40%:20%, respectively. Lower case letters indicate that the rice quality characteristic are significantly different among the treatments ($p < 0.05$, LSD method). BR: brown rice rate; MR: milled rice rate; HMR: head rice rate; AC: amylose content; PC: protein content.

### 3.2.2. Appearance Quality

The rice chalkiness rate, chalkiness, aspect ratio, appearance, hardness, and eating quality (such as taste and eating value) were the highest in 2019 (Table 4). The hardness, taste, and taste values of the wheat season were the highest, followed by the rape season; the aspect ratio was the highest in the rape season, followed by the wheat season. The previous crop had a significant impact on the appearance and taste quality of rice. Appearance (chalkiness rate, aspect ratio, appearance, hardness) and taste quality (taste and eating value) were highest under different nitrogen application rates in the rapeseed season and wheat season. The chalkiness is based on the rapeseed or wheat season. Conventional nitrogen fertilization was the highest in the rape season and wheat season. Different nitrogen application rates have a great impact on the appearance and taste quality of rice. Different nitrogen fertilizer management had a great impact on the chalkiness rate,

chalkiness size, and chalkiness of rice appearance. Chalkiness and chalkiness rate were M0 > M3 > M2 > M1 under different nitrogen fertilizer operations; aspect ratio was M0 > M3 > M2 > M1 under different nitrogen fertilizer operations. Appearance under different nitrogen fertilizer operations was M0 > M2 > M3 > M1; under the oil–rice rotation, the rice season M3 treatment increased the rice length-to-width ratio, but at the same time, increased the rice chalkiness and reduced the appearance quality of the rice. Under the wheat–rice rotation, the M2 treatment in the rice season increased the aspect ratio of the rice, and at the same time, increased the chalkiness of the rice and also reduced the appearance quality of the rice. The rice taste and mouthfeel under different nitrogen fertilizer managements were M0 > M3 > M2 > M1. Among them, rice taste M3 increased 0.35% and 2.43% compared with M2 and M1, respectively. Nitrogen fertilizer management under the oil/wheat–rice rotation had a great impact on the eating quality. The taste of rice under the oil–rice rotation increased with the increase in the ratio of the base tiller fertilizer to the total nitrogen application, and the M3 operation was the best treatment. Under the wheat–rice rotation, the taste of rice increased first and then later changed with the increase in the ratio of the base tiller fertilizer to the total nitrogen application. The M2 operation is the best. Therefore, it is more appropriate to reduce the application ratio of ear fertilizer, which is then conducive to improving the eating quality of rice.

### 3.2.3. RVA Profile Characteristic Value of Rice

The characteristic of the starch RVA profile is an important indicator of the taste of rice. Generally speaking, varieties with better eating quality generally have a larger disintegration value and lower cut-off value. The peak viscosity, hot paste viscosity, cold glue viscosity, and disintegration value were the largest in 2019, while the peak time, reduction value, and gelatinization temperature were the highest in 2018 (Table 5). The gum viscosity and disintegration values were highest in the wheat season, followed by the rape season, while the peak time, reduction value, and gelatinization temperature were the highest in the rape season, followed by the wheat season. Different previous crops will affect the RVA of rice starch. The effect of nitrogen application rate on the RVA spectral characteristics of starch is clear. Peak viscosity, hot paste viscosity, cold glue viscosity, disintegration value, and peak time were higher under conventional nitrogen application, while the reduction value and gelatinization temperature were the largest under reduced nitrogen application. Different nitrogen fertilizer operations have a great impact on the characteristic value of the RVA profile of rice starch. The peak viscosity and hot slurry viscosity under different nitrogen fertilizer operations was M0 > M1 > M2 > M3; the disintegration value under different nitrogen fertilizer operations was M3 > M2 > M1 > M0; M3 increased relative to M2, M1, and M0 by 2.54%, 2.88%, and 6.98%, respectively. The cold glue viscosity and gelatinization temperature under different nitrogen fertilizer operations was expressed as M0 > M1 > M2 > M3; the reduction value was M1 > M3 > M2 > M0; M1 relative to M3, M2 and M0 increased by 27.13%, 52.62%, and 71.04%, respectively. The peak viscosity, hot pulp viscosity, disintegration value, and cold gel viscosity of rice under different nitrogen fertilizer management treatments under the oil–rice rotation (except for the control treatment without nitrogen fertilizer; M0) were the highest and were the same as the base tiller fertilizer. The proportion of nitrogen increased gradually, while the change in the reduction value was opposite; under the wheat–rice rotation, the proportion of base tiller fertilizer in the total nitrogen application increased first and then decreased. The difference between the treatments was significant, and the disintegration value was the largest under the M2 treatment. The reduction value was the smallest at M2. It shows that reasonable nitrogen fertilizer management under the rapeseed/wheat–rice rotation is beneficial in improving the eating quality of rice.

**Table 4.** The effects of the N application rate in the previous season and N management in the rice season on rice appearance quality and eating quality.

| Year | Treatment | | | CP (%) | CD (%) | L/W | Appearance | Comprehensive | Hardness | Mouthfeel |
|---|---|---|---|---|---|---|---|---|---|---|
| 2018 | Pr | Nc | M0 | 57.06 c | 20.26 b | 2.50 cd | 8.43 cd | 85.33 cd | 3.36 h | 7.56 def |
| | | | M1 | 35.92 o | 11.68 klm | 2.50 cd | 7.80 ijk | 74.33 g | 3.96 bc | 6.30 m |
| | | | M2 | 35.57 o | 11.26 lm | 2.50 cd | 7.63 jk | 77.66 f | 4.13 abc | 6.73 jkl |
| | | | M3 | 38.36 lmn | 12.26 jkl | 2.60 a | 7.23 m | 80.00 e | 3.86 cd | 6.90 ijk |
| | | | average | 41.73 | 13.87 | 2.53 | 7.77 | 79.33 | 3.83 | 6.8725 |
| | | Nr | M0 | 53.80 d | 17.21 cdef | 2.50 cd | 8.50 cd | 85.33 cd | 3.50 gh | 7.60 def |
| | | | M1 | 33.23 p | 10.63 m | 2.56 ab | 7.60 jk | 76.33 fg | 4.33 ab | 6.50 lm |
| | | | M2 | 35.52 o | 11.07 lm | 2.52 cd | 7.67 jk | 77.33 f | 4.10 abc | 6.66 kl |
| | | | M3 | 37.03 mno | 11.23 lm | 2.61 a | 7.30 lm | 77.66 f | 3.66 fg | 6.70 jkl |
| | | | average | 39.90 | 12.54 | 2.55 | 7.77 | 79.16 | 3.90 | 6.865 |
| | Pw | Nc | M0 | 62.63 a | 23.03 a | 2.60 a | 8.83 ab | 88.33 ab | 4.13 abc | 8.33 b |
| | | | M1 | 55.76 cd | 20.61 b | 2.43 e | 8.43 cd | 84.00 d | 3.50 gh | 7.70 def |
| | | | M2 | 50.50 ef | 17.60 cde | 2.51 cd | 8.36 de | 84.66 cd | 3.63 fg | 7.76 de |
| | | | M3 | 45.96 hi | 16.66 def | 2.46 de | 8.40 de | 84.66 cd | 4.16 abc | 7.63 def |
| | | | average | 53.71 | 19.48 | 2.50 | 8.51 | 85.41 | 3.86 | 7.855 |
| | | Nr | M0 | 60.43 b | 22.36 a | 2.61 a | 9.01 a | 89.33 a | 3.30 h | 8.70 a |
| | | | M1 | 43.26 k | 14.66 hi | 2.51 cd | 8.56 bc | 84.33 d | 3.60 gh | 7.63 def |
| | | | M2 | 43.70 jk | 15.86 fgh | 2.50 cd | 8.36 de | 86.33 bcd | 4.36 a | 7.90 cd |
| | | | M3 | 49.76 fg | 18.05 cd | 2.50 cd | 8.53 bc | 86.00 bcd | 4.23 ab | 7.86 cd |
| | | | average | 49.29 | 17.73 | 2.53 | 8.62 | 86.50 | 3.87 | 8.0225 |
| 2019 | Pr | Nc | M0 | 55.70 cd | 18.54 c | 2.61 a | 8.66 bc | 86.00 bcd | 3.53 gh | 7.86 cd |
| | | | M1 | 38.83 lm | 12.74 jk | 2.51 cd | 7.76 ijk | 77.00 f | 4.16 abc | 6.63 klm |
| | | | M2 | 38.56 lm | 13.46 ij | 2.50 cd | 8.03 hi | 76.66 f | 3.83 de | 6.71 jkl |
| | | | M3 | 40.15 l | 12.87 jk | 2.56 ab | 7.53 kl | 81.00 e | 4.20 ab | 7.16 ghi |
| | | | average | 43.31 | 14.40 | 2.55 | 8.00 | 80.17 | 3.93 | 7.09 |
| | | Nr | M0 | 55.85 cd | 18.26 c | 2.60 a | 8.26 fg | 84.00 d | 3.56 gh | 7.41 efg |
| | | | M1 | 35.51 o | 11.76 klm | 2.50 cd | 8.26 fg | 80.00 e | 3.80 ef | 7.03 hij |
| | | | M2 | 38.76 lm | 12.35 jkl | 2.50 cd | 8.03 hi | 80.33 e | 4.20 ab | 7.10 ghi |
| | | | M3 | 49.23 fg | 16.33 efg | 2.56 ab | 7.86 ij | 83.66 d | 3.53 gh | 7.36 fgh |
| | | | average | 44.84 | 14.68 | 2.54 | 8.10 | 82.00 | 3.77 | 7.225 |
| | Pw | Nc | M0 | 63.43 a | 23.03 a | 2.62 a | 8.83 ab | 88.33 ab | 4.16 abc | 8.33 b |
| | | | M1 | 36.50 no | 12.97 jk | 2.50 cd | 8.43 cd | 83.66 d | 3.80 ef | 7.60 def |
| | | | M2 | 45.37 ij | 15.16 gh | 2.53 bc | 8.20 gh | 86.33 bcd | 3.50 gh | 7.90 cd |
| | | | M3 | 44.06 ijk | 14.66 hi | 2.50 cd | 8.61 bc | 84.33 d | 3.40 h | 7.40 efg |
| | | | average | 47.34 | 16.46 | 2.54 | 8.52 | 85.66 | 3.72 | 7.8075 |
| | | Nr | M0 | 63.03 a | 22.36 a | 2.61 a | 9.01 a | 89.33 a | 3.30 h | 8.71 a |
| | | | M1 | 47.74 gh | 17.24 cdef | 2.50 cd | 8.73 ab | 84.33 d | 3.56 gh | 7.56 def |
| | | | M2 | 51.85 e | 18.21 c | 2.50 cd | 8.33 ef | 87.00 abc | 3.66 fg | 8.13 bc |
| | | | M3 | 49.57 fg | 18.19 c | 2.50 cd | 8.50 cd | 85.33 cd | 4.33 ab | 7.66 def |
| | | | average | 53.05 | 19.00 | 2.53 | 8.64 | 86.50 | 3.71 | 8.015 |

Pr represents rapeseed; Nc and Nr represent the conventional nitrogen application and reduced nitrogen application in rape season, respectively; Pw represents wheat; Nc and Nr represent the conventional nitrogen application and reduced nitrogen application in the wheat season, respectively. M0 represents zero N was used in rice season; M1, M2 and M3 represent based on an application rate of 150 kg/hm2 N in the rice season, three N management models were applied, in which the application ratio of base:tiller:panicle fertilizer was 20%:20%:60%, 30%:30%:40%, and 40%:40%:20%, respectively. Lower case letters indicate that the rice appearance quality and eating quality are significantly different among the treatments ($p < 0.05$, LSD method). CP: chalk grain rate; CD: chalkiness degree; L/W: Grain length/Width ratio.

**Table 5.** The effects of the N application rate in the previous season and N management in the rice season on the RVA profile characteristic value of rice.

| Year | Treatment | | | PV (cp) | HV (cp) | BD (cp) | CV (cp) | SB (cp) | Pt (cp) | PT (cp) |
|---|---|---|---|---|---|---|---|---|---|---|
| 2018 | Pr | Nc | M0 | 305.55 ij | 200.72 cde | 104.77 jkl | 317.61 ef | 16.10 cdef | 6.13 abc | 78.45 a |
| | | | M1 | 281.41 o | 161.65 op | 96.14 m | 301.39 lm | 19.50 bc | 6.11 abc | 78.13 ab |
| | | | M2 | 289.77 lmn | 186.30 ij | 96.11 m | 308.36 ij | 22.63 ab | 6.18 ab | 78.35 ab |
| | | | M3 | 289.97 lmn | 193.91 efg | 106.94 hi | 312.66 gh | 16.47 cdef | 6.07 abc | 77.28 bc |
| | | | average | 291.68 | 185.65 | 100.96 | 310.01 | 18.68 | 6.12 | 78.05 |
| | | Nr | M0 | 300.35 jk | 207.02 abc | 104.30 klm | 311.22 gh | 11.66 hij | 6.09 abc | 78.15 ab |
| | | | M1 | 284.27 no | 165.58 nop | 94.53 n | 309.66 hi | 18.41 cd | 6.15 ab | 77.61 abc |
| | | | M2 | 283.95 no | 180.13 kl | 108.97 gh | 296.61 m | 22.50 ab | 5.95 def | 77.41 abc |
| | | | M3 | 285.67 no | 192.85 efg | 114.97 fg | 303.43 kl | 17.64 cde | 6.02 abc | 78.26 ab |
| | | | average | 288.56 | 186.40 | 105.66 | 305.23 | 17.55 | 6.05 | 77.86 |

**Table 5.** *Cont.*

| Year | Treatment | | | PV (cp) | HV (cp) | BD (cp) | CV (cp) | SB (cp) | Pt (cp) | PT (cp) |
|---|---|---|---|---|---|---|---|---|---|---|
| | | | M0 | 342.45 b | 200.31 cde | 129.43 abc | 335.67 b | −2.25 n | 6.02 abc | 76.62 ghi |
| | | | M1 | 296.60 kl | 172.44 mn | 139.01 a | 331.01 bc | 4.39 lm | 6.01 bcd | 76.88 fg |
| | | Nc | M2 | 328.03 de | 196.83 def | 124.69 cd | 315.11 fg | 3.36 lm | 5.95 cde | 76.81 fg |
| | | | M3 | 307.52 hi | 192.59 fgh | 121.89 cd | 304.58 jkl | 12.47 ghi | 5.97 cde | 76.68 ghi |
| | Pw | | average | 318.65 | 190.54 | 128.70 | 321.59 | 4.49 | 5.99 | 76.75 |
| | | | M0 | 337.66 bc | 202.43 bcd | 126.45 bc | 337.33 ab | −2.107 n | 5.94 def | 76.58 ghi |
| | | | M1 | 316.38 fg | 197.58 def | 122.14 cd | 322.58 def | 6.57 kl | 6.03 bcd | 76.49 ghi |
| | | Nr | M2 | 321.30 ef | 197.97 def | 126.72 bc | 321.53 def | 6.32 kl | 6.13 abc | 76.95 ef |
| | | | M3 | 313.94 gh | 189.64 ghi | 117.47 def | 319.57 def | 10.25 ij | 6.03 bcd | 77.13 de |
| | | | average | 322.32 | 196.91 | 123.15 | 325.25 | 5.26 | 6.03 | 76.79 |
| | | | M0 | 298.65 jk | 195.38 defg | 108.08 ghi | 313.14 gh | 15.30 defg | 6.09 abc | 78.31 ab |
| | | | M1 | 271.52 p | 162.86 op | 106.52 ij | 314.50 fg | 24.27 a | 6.08 abc | 78.13 ab |
| | | Nc | M2 | 285.47 no | 172.19 mn | 108.91 gh | 294.28 n | 25.08 a | 5.95 def | 77.63 abc |
| | | | M3 | 296.69 kl | 188.61 hi | 115.80 efg | 296.69 m | 15.39 defg | 6.02 abc | 78.06 abc |
| | Pr | | average | 288.08 | 179.76 | 109.80 | 304.65 | 20.01 | 6.04 | 78.03 |
| | | | M0 | 295.60 klm | 186.47 ij | 115.05 fg | 312.72 gh | 13.01 fghi | 6.06 abc | 78.13 ab |
| | | | M1 | 284.15 no | 167.52 mno | 102.27 lm | 306.77 ijk | 26.08 a | 6.09 abc | 77.86 abc |
| | | Nr | M2 | 284.83 no | 180.68 kl | 114.55 fg | 304.25 jkl | 16.72 cdef | 6.02 abc | 78.21 ab |
| | | | M3 | 288.80 mno | 174.64 lm | 116.94 def | 303.28 kl | 16.69 cdef | 5.95 def | 77.83 abc |
| 2019 | | | average | 288.35 | 177.33 | 112.15 | 306.76 | 18.13 | 6.03 | 78.01 |
| | | | M0 | 324.85 e | 208.75 ab | 131.17 abc | 326.78 cd | 3.39 lm | 5.94 def | 76.25 i |
| | | | M1 | 313.62 gh | 159.25 p | 108.27 ghi | 324.22 cde | 13.94 efgh | 6.15 ab | 77.18 cd |
| | | Nc | M2 | 314.62 fg | 194.98 defg | 124.08 cd | 284.33 o | 1.56 m | 5.79 f | 77.45 abc |
| | | | M3 | 265.56 p | 186.52 ij | 116.19 def | 314.89 fg | 5.19 klm | 6.01 bcd | 76.93 ef |
| | Pw | | average | 304.66 | 187.38 | 119.85 | 312.56 | 6.02 | 5.97 | 76.95 |
| | | | M0 | 359.51 a | 210.89 a | 104.96 jk | 344.87 a | −9.117 o | 6.22 a | 77.31 abc |
| | | | M1 | 304.61 ij | 183.34 jk | 118.66 de | 306.47 ijk | 8.52 jk | 5.99 bcd | 77.45 abc |
| | | Nr | M2 | 332.61 cd | 196.66 def | 135.05 ab | 343.87 a | 4.25 lm | 5.91 ef | 76.32 hi |
| | | | M3 | 325.36 e | 195.40 defg | 126.19 bcd | 322.94 def | 11.55 hij | 6.09 abc | 77.39 abc |
| | | | average | 330.52 | 196.57 | 121.15 | 329.54 | 3.80 | 6.05 | 77.12 |

Pr represents rapeseed; Nc and Nr represent the conventional nitrogen application and reduced nitrogen application in rape season, respectively; Pw represents wheat; Nc and Nr represent the conventional nitrogen application and reduced nitrogen application in the wheat season, respectively. M0 represents zero N was used in rice season; M1, M2 and M3 represent based on an application rate of 150 kg/hm$^2$ N in the rice season, three N management models were applied, in which the application ratio of base:tiller:panicle fertilizer was 20%:20%:60%, 30%:30%:40%, and 40%:40%:20%, respectively. Lower case letters indicate that the RVA profile characteristic value of rice are significantly different among the treatments ($p < 0.05$, LSD method). PV: peak viscosity; HV: hot viscosity; BD: break disintegration; CV: cool viscosity; SB: setback; PT: peak time.

## 4. Discussion

### 4.1. Effects of Nitrogen Fertilizer Management on Yield in Different Rotation Modes

How to increase crop yields and reduce nitrogen fertilizer input to increase the efficient absorption and utilization of nitrogen fertilizer by crops is one of the current hot spots in the domain of agricultural research. Existing studies have shown that straw return to the field, nitrogen fertilizer management, and straw return to the field combined with nitrogen fertilizer have important regulatory effects on rice efficiency and yield, carbon and nitrogen metabolism, and high-efficiency utilization of nitrogen [16]. A large number of studies have shown that appropriate nitrogen fertilizer management, nitrogen application rate [17], and straw return to the field and nitrogen fertilizer management [18] can promote a significant increase in the cumulative amount of nitrogen uptake by rice at the mature stage, which can greatly reduce the amount of nitrogen fertilizer applied. Studies have shown that reduced fertilization has little effect on yield in crop rotation systems such as wheat–rice, rapeseed–rice, and corn-cole [19,20]. Zhang Weile et al. showed that under the condition of returning straw to the field, the nitrogen demand of crops could be met by the post-fertilization of nitrogen fertilizer, and high and stable crop yields can be ensured [21]. Yanfengjun et al. [22] reported that when the ratio of base tiller fertilizer to ear fertilizer was 6:4, high yields were ensured. This study believes that whether rice yield increases significantly under the rapeseed/wheat–rice rotation is closely related to the ratio of base tiller fertilizer to panicle fertilizer nitrogen fertilizer. The results of this study show that in the rapeseed–rice planting system, the yield of rice under different treatments depends on the reduction of nitrogen in the rape season, which is the largest when combined with

the nitrogen fertilizer M3 in the rice season. The advantages of the huge root system of rapeseed with a large accumulation of nutrients, large biomass of straw returned to the field, and high nutrient release efficiency may be the main reasons for its significant role in promoting rice production in the rice season. In the wheat–rice rotation system, rice yield under different treatments is represented by conventional nitrogen application in the wheat season, which is highest when combined with the nitrogen fertilizer M2 operation in the rice season. It is possible that the reasonable management of nitrogen fertilizer in the wheat–rice rotation system reduces the nitrogen accumulation in the early stage of rice growth, inhibits the occurrence of ineffective tillers, and then meets the nitrogen demand during the grain development process through top dressing, ensuring the production of rice. In different crop rotation systems, reasonable nitrogen fertilizer management can better coordinate straw rot and rice growth to compete for nitrogen, ensure early and stable rice tillers, achieve the expected number of panicles, and coordinate the contradiction between foot panicles and large panicles to achieve high yields. Nitrogen fertilizer management under other crop rotation modes and the background value of soil nutrients will affect nitrogen fertilizer management. The effect of returning straw into the field on the formation of rice yield remains to be further studied in this respect.

*4.2. Effects of Nitrogen Fertilizer Management on Rice Processing, Appearance, RVA and Nutritional Quality under Different Rotation Models*

The types of straw and the amount of nitrogen applied have substantial effects on rice processing, appearance quality, and nutritional quality. After returning the straw to the field, it can reduce the chalkiness rate, the size and the degree of chalkiness, increase the brown rice rate, the polished rice rate, and the whole rice rate, and improve the processing quality and appearance quality of rice [23,24]. Previous research reveals that an appropriate amount of N fertilizer can decrease the chalky kernel rate, while the overuse of N can increase the chalky kernel rate and undesirable grain appearance [25,26]. The degree of chalkiness was significantly negatively related to eating/cooking quality. This was primarily due to the fact that high chalkiness implies a low density of starch granules, and therefore, the grains are more prone to breakage during cooking [27]. The amylose content was decreased with the increasing nitrogen level. According to a previous study, there are A- and B-types of starch granules in the endosperm [28]. This study shows that under different rotation modes, returning crop stalks to the field had significant or effects on the rice milling rate, chalkiness, hardness, and protein content of hybrid indica rice. We believe that the effect of straw return on rice chalkiness may be mainly related to the nitrogen and carbon supply of grain filling and the dynamic changes in grain filling [29]. The mechanism of returning all straws to the field to improve the appearance of rice needs to be further explored. The protein of rice is an ideal plant protein, which is easily absorbed by the human body and is the main indicator of the nutritional quality of rice. Reasonable nitrogen fertilizer management can substantially improve the quality of rice [30]. Wopereis-Pura et al. [31] researched that more panicle fertilizer application can significantly improve the processing quality of rice. The results of this study also show that as the percentage of panicle fertilizer in the total nitrogen application increases, the brown rice rate, polished rice rate, and whole rice rate increase. Increasing the ratio of panicle fertilizer to the total nitrogen application can significantly improve the processing quality of rice. Marwanto et al. [32] believed that an increase in the proportion of nitrogen fertilizer does not increase the chalkiness rate of rice, but the chalkiness becomes larger. The results of the current study are consistent with this. It shows that increasing the ratio of ear fertilizer to the total nitrogen application can significantly improve the nutritional quality of machine-grown, high-quality edible rice under a rapeseed–rice rotation system. The effect of returning the whole amount of straw to the field on the eating quality of rice is still lacking. Starch RVA profile characteristics are important indicators for evaluating rice quality and are closely related to cooking and eating quality. After the straw is returned to the field, the maximum viscosity and disintegration value both increase, but the reduction value becomes smaller [33]. In this study, the reduced amount of nitrogen fertilization in

the rape season under the rapeseed–rice rotation and the eating quality of the rice under the treatment of the M3 operation in the rice season are the best, and the rice taste quality was best following the conventional nitrogen application in the wheat season combined with the M2 operation in the rice season under wheat–rice rotation. This shows that reasonable nitrogen fertilizer management under the rapeseed/wheat–rice rotation is beneficial in improving the eating quality of rice.

## 5. Conclusions

Optimizing nitrogen fertilizer management can increase rice yield and rice quality under rapeseed/wheat–rice rotation systems. Reduced N for rapeseed and the panicle fertilizer of 40%:40%:20% in rice season under a rapeseed–rice rotation system can be recommended to stabilize yield and high-quality rice production and can be used as an N-saving and environmentally friendly measure in rapeseed–rice rotation systems in southern China.

**Author Contributions:** P.M. and Y.L. are co-first authors. P.M.: Investigation, Methodology, Writing—original draft. Y.L.: Resources, Software, Writing—original draft. X.L.: Data curation. P.F.: Investigation, Methodology. Z.Y.: Methodology. Y.S.: Software. R.Z. and J.M.: Conceptualization, Funding acquisition, Supervision, Validation. All authors have read and agreed to the published version of the manuscript.

**Funding:** This study was supported by the National Key Research and Development Program of China (Nos. 2017YFD0301701; 2017YFD0301706) and the Scientific Research Fund of Sichuan Provincial Education Department (18ZA0390).

**Institutional Review Board Statement:** Not applicable.

**Informed Consent Statement:** Informed consent was obtained from all subjects involved in the study.

**Data Availability Statement:** The data presented in this study are available on request from the authors.

**Acknowledgments:** We would like to thank our teacher for carefully reading and correcting our manuscript and providing technical assistance and financial support for the study, as well as our scientific research team for their contribution to this paper.

**Conflicts of Interest:** The authors declare that they have no known competing financial interests or personal relationships that could appear to influence the work reported in this paper.

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
