# Peer review of "Reasonable Nitrogen Fertilizer Management Improves Rice Yield and Quality under a Rapeseed/Wheat–Rice Rotation System"

_agriculture, doi:10.3390/agriculture11060490_

Round 1
Reviewer 1 Report
Dear Author
The manuscript ‘Reasonable nitrogen fertilizer management improves rice yield and quality under a rapeseed/wheat-rice rotation system’ concerns an important agronomic topic. It is written clearly and concisely. The presented Conclusions correspond to the results obtained from the statistical analyzes. However, there are some inaccuracies that need to be corrected.
Materials and methods
A more detailed explanation is required in the 'methods' section.
- Unclear is the last part of sentence ‘The second factor (two levels) was the N application rate, with a conventional N application (Nc; 180 kg/hm2 N in rape season, 150 kg/hm2 N in wheat season), reduced N (Nr; 150 kg/hm2 N in rape season, 120 kg/hm2 N in wheat season), and N fertilizer as basal manure and top dressing at a 5:5 ratio.’(Page 3, lines 2-4)
- In the sentence ‘The weeds were controlled in the oilseed, rape, and rice plots…’ no comma is needed after 'oilseed'. (Page 3, line 30)
- Quantities of active substances of pesticides per ha should be provided. (Page 3, lines 30-32)
- The sentence ‘At maturity stage, all rice plants were selected from each plot to test the rice yield (GY) were calculated according to the actual number of harvested plants; the value was adjusted to 13.5% moisture. ‘ is not fully understood. (Page 3, lines 35-36)
Results
- Correct the sentence, from 'Under different operations, the performance was: M3 > M2 > M1 > M0. ' to 'Under different nitrogen operations, the performance was: M3 > M2 > M1 > M0’.
- The sentence ‘M3 was relative to M2, M1, and M0, and increased by 1.39%, 4.61%, and 55.67%, respectively’ is not complete. Rewrite it. (Page 4)
- Each table and each figure must be self-explanatory. Explain precisely what the abbreviations mean (Nc, Nr, M0, M1, M2, M3) under figure 2 and under the tables 2, 3, 4, and 5.
Conclusions
- In this chapter, try not to use abbreviations. Describe exactly what 'reduced N for rapeseed' and 'M3 N regime in rice' means.
General comments
- SI Units (International System of Units) should be used
- Use the word ‘rapeseed’ instead ‘oilseed rape, oil e.t.c.)’ throughout the manuscript.
- The words 'ear' and 'panicle' are used interchangeably. Use only the word 'panicle' for rice throughout the manuscript.
Author Response
Response to Reviewer 1 Comments
Point 1: Unclear is the last part of sentence ‘The second factor (two levels) was the N application rate, with a conventional N application (Nc; 180 kg/hm2 N in rape season, 150 kg/hm2 N in wheat season), reduced N (Nr; 150 kg/hm2 N in rape season, 120 kg/hm2 N in wheat season), and N fertilizer as basal manure and top dressing at a 5:5 ratio.’(Page 3, lines 2-4)
Response 1: Thank you for your comment. We have already explained clearly(line84-85).
Point 2: In the sentence ‘The weeds were controlled in the oilseed, rape, and rice plots…’ no comma is needed after 'oilseed'. (Page 3, line 30)
Response 2: Thank you for your comment. We have revised it(line112)
Point 3: 3.Quantities of active substances of pesticides per ha should be provided. (Page 3, lines 30-32)
Response 3: Thank you for your comment. We have revised it(line112-114)
Point 4:4.The sentence ‘At maturity stage, all rice plants were selected from each plot to test the rice yield (GY) were calculated according to the actual number of harvested plants; the value was adjusted to 13.5% moisture. ‘ is not fully understood. (Page 3, lines 35-36)
Response 4: Thank you for your comment. We have revised it(line119)
Point 5: 1.Correct the sentence, from 'Under different operations, the performance was: M3 > M2 > M1 > M0. ' to 'Under different nitrogen operations, the performance was: M3 > M2 > M1 > M0’.The sentence ‘M3 was relative to M2, M1, and M0, and increased by 1.39%, 4.61%, and 55.67%, respectively’ is not complete. Rewrite it. (Page 4)
Response 5: Thank you for your comment. We have corrected it in the revised manuscript(line 170).
Point 6: Each table and each figure must be self-explanatory. Explain precisely what the abbreviations mean (Nc, Nr, M0, M1, M2, M3) under figure 2 and under the tables 2, 3, 4, and 5.
Response 6: Thank you for your comment. We have revised it (line 186-193,)
Point 7: In this chapter, try not to use abbreviations. Describe exactly what 'reduced N for rapeseed' and 'M3 N regime in rice' means.
Response 7: Thank you for your comment.We have revised it
Point 8:1.SI Units (International System of Units) should be used. Use the word ‘rapeseed’ instead ‘oilseed rape, oil e.t.c.)’ throughout the manuscript.The words 'ear' and 'panicle' are used interchangeably. Use only the word 'panicle' for rice throughout the manuscript.
Response 8: Thank you for your comment.We have revised it.

Reviewer 2 Report
In the manuscript “Reasonable nitrogen fertilizer management improves rice yield and quality under a rapeseed/wheat-rice rotation system« authors Peng Ma, Yan Lan, Xu Lv, Ping Fan, Zhiyuan Yang , Yongjian Sun, Rongping Zhang, Jun Ma discussed about optimizing nitrogen fertilizer management that can increase rice yield and rice quality under rapeseed/wheat-rice rotation systems.
Abstract Please avoid abbreviations.
respectively. a delete full stop
Key words ;; use right punctations!
Introduction
Is informative.
…rice [2,3].Reasonable…; rice [13].Under t…; of rice [15].The above; Spaces are missing
…be used,For…; used [14]Under conditions… punctations!
Materials and Methods
Clear and well explained
H2SO4 and HClO4 subscriptions (H2SO4)!
Results
Table 1: Y:year;P: previous crop;… Puncations!
M0,and… space is missing
significant(Table2). I… space is missing
Table2 space is missing
Table 2: Y:year;P: previous crop; N: nitrogen rate;M:nitrogen management. BR:brown rice rate;MR:milled rice rate;HMR:head rice rate;CP:chalk grain rate;CD:chalkness degree;AC:amylose content;PC:protein content;PV:peak viscosity;HV:hot viscosity;BD:break disintegration;CV:cool viscosity;SB:setback;PT:peak time. Spaces are missing
Table 3, BR:brown rice rate;MR:milled rice rate;HMR:head rice rate;AC:amylose content;PC:protein content. Spaces are missing
e appearance.Chalkines Space is missing
Table 4 CP:chalk grain rate;CD:chalkness degree;L/W: Grain length/Width ratio.. Spaces are missing
Table5 Space is missing
Table 5 PV:peak viscosity;HV:hot viscosity;BD:break disintegration;CV:cool viscosity;SB:setback;PT:peak time…. Spaces are missing
Discussion
7∶ 3 space?
The first paragraph without results from the present study is too long, and the second part is without references. Own result and literature data should be more connected.
Conclusions
Ok. I suggest avoid of abbreviations.
Specific comments
I would like to highlight the hard and complex work of the authors. Generally speaking, the text is clearly written, and clearly describes experiments. Discussion is sound and relevant.
My suggestions: minor revision
Author Response
Point 1: Please avoid abbreviations.respectively. a delete full stop.;; use right punctations!
Response 1: Thank you for your comment. We have revised it(line22-26,28).
Point 2: Is informative.…rice [2,3].Reasonable…; rice [13].Under t…; of rice [15].The above; Spaces are missing…be used,For…; used [14]Under conditions… punctations!
Response 2: Thank you for your comment.We have revised it.(line36,53,55,57,59)
Point 3: Clear and well explained H2SO4 and HClO4 subscriptions (H2SO4)!
Response 3: Thank you for your comment. We have revised it(line75)
Point 4:Table 1: Y:year;P: previous crop;… Puncations!M0,and… space is missing significant(Table2). I… space is missing.Table2 space is missing.Table 2: Y:year;P: previous crop; N: nitrogen rate;M:nitrogen management. BR:brown rice rate;MR:milled rice rate;HMR:head rice rate;CP:chalk grain rate;CD:chalkness degree;AC:amylose content;PC:protein content;PV:peak viscosity;HV:hot viscosity;BD:break disintegration;CV:cool viscosity;SB:setback;PT:peak time. Spaces are missing.Table 3, BR:brown rice rate;MR:milled rice rate;HMR:head rice rate;AC:amylose content;PC:protein content. Spaces are missinge appearance.Chalkines Space is missing.Table 4 CP:chalk grain rate;CD:chalkness degree;L/W: Grain length/Width ratio.. Spaces are missing.Table5 Space is missing.Table 5 PV:peak viscosity;HV:hot viscosity;BD:break disintegration;CV:cool viscosity;SB:setback;PT:peak time…. Spaces are missing
Response 4: Thank you for your comment.We have revised it.
Point 5: 7∶ 3 space?The first paragraph without results from the present study is too long, and the second part is without references. Own result and literature data should be more connected.
Response 5: Thank you for your comment. We have revised it.
Point 5: The figure numbering is incorrect. The statistically significant differences are not presented in Table 4.Conclusions could be more extensive.
Response 5: Thank you for your comment. We have revised the number of this paper figure and added the statistically significant differences,and the conclusions of this paper was revised.
